Co-father relationships among the Suruí (Paiter) of Brazil

Walker Robert S. 1 walkerro@missouri.edu
Yvinec Cédric 2
Ellsworth Ryan M. 1
Bailey Drew H. 3
1 Department of Anthropology, University of Missouri , Columbia, MI , USA
2 Fondation Thiers, Laboratoire d’Anthropologie Sociale , Paris , France
3 School of Education, University of California , Irvine, CA , USA
Wagner Jennifer
Electronic publication date: 2015 Apr 14
Publication date: 2015
Volume: 3
Electronic Location ID: e899
Received 2015 Feb 18; Accepted 2015 Mar 29
Copyright: © 2015 Walker et al.
Copyright year: 2015
Copyright holder: Walker et al.
License: This is an open access article distributed under the terms of the Creative Commons Attribution License, which permits unrestricted use, distribution, reproduction and adaptation in any medium and for any purpose provided that it is properly attributed. For attribution, the original author(s), title, publication source (PeerJ) and either DOI or URL of the article must be cited.
License URL: https://creativecommons.org/licenses/by/4.0/

Keywords: Multiple fathers, Partible paternity, Reproductive strategies, Cooperative breeding, Amazonia

Funding: EHESS (Paris), Collège de France, and Fondation Fyssen National Geographic Society Research and Exploration #9165-12 Ethnographic data were collected by Yvinec in 2005–2007 and 2013 with financial support from the EHESS (Paris), Collège de France, and Fondation Fyssen. Analysis of the data and interpretation of the results were conducted by Walker, Ellsworth, and Bailey with financial support provided by a National Geographic Society Research and Exploration grant (#9165-12). The funders had no role in study design, data collection and analysis, decision to publish, or preparation of the manuscript.

==============================
Partible paternity refers to the conception belief that children can have multiple fathers (“co-fathers”) and is common to indigenous cultures of lowland South America. The nature of social relationships observed between co-fathers reveals information about the reproductive strategies underlying partible paternity. Here we analyze clan, genealogical, and social relationships between co-fathers for the Suruí, an indigenous horticultural population in Brazil. We show that co-fathers roughly assort into two separate categories. In the affiliative category, co-father relationships are amicable when they are between close kin, namely brothers and father-son. In the competitive category, relationships are more likely of avoidance or open hostility when between more distant kin such as cousins or unrelated men of different clans. Results therefore imply multiple types of relationships, including both cooperative and competitive contexts, under the rubric of partible paternity. These complexities of partible paternity institutions add to our knowledge of the full range of cross-cultural variation in human mating/marriage arrangements and speak to the debate on whether or not humans should be classified as cooperative breeders.

Introduction

Partible paternity refers to the concept that children can have more than one genitor (Beckerman et al., 1998). In contrast to the realities of sexual reproduction, conception under partible paternity is thought to be a cumulative process that involves seminal inputs from multiple men in the production of offspring. Such an outlook on reproduction is accompanied by polygynandrous mating and institutionalized forms of extramarital relationships in addition to marital bonds (Beckerman & Valentine, 2002). Intriguingly, partible paternity appears almost exclusively in lowland South America where it is nearly ubiquitous in the Arawá, Carib, Macro-Jê, Pano, and Tupi language families (Walker, Flinn & Hill, 2010). At last count, we know of 61 societies across Greater Amazonia with traditional beliefs in partible paternity and only 24 with singular paternity beliefs. Ethnographic descriptions of partible and singular paternity cultures suggest important differences in sociosexual dynamics between these two categories of societies, particularly in the degree to which female sexual autonomy and extramarital relationships are tolerated (Beckerman & Valentine, 2002; Walker, Flinn & Hill, 2010).

Partible paternity has been used as an example of cooperative breeding (Hrdy, 2000; Hrdy, 2009). Cooperative breeding is a social system in which individuals help care for offspring that are not their own at the expense of their own direct reproduction (Emlen, 1991; Lukas & Clutton-Brock, 2012). This definition is often extended to include care from all non-maternal helpers, including putative fathers (Hrdy, 2000; Hrdy, 2009), sub-adults (Kramer, 2005), and grandparents (Hawkes, O’Connell & Blurton Jones, 1997) even in systems like humans with low reproductive skew. While there is some disagreement over the appropriate term (Bogin, Bragg & Kuzawa, 2014), extensive cooperation and alloparental care in humans has led a number of authors to espouse cooperative breeding in the broad sense as an apt description of human systems (Mace & Sear, 2005; Hill & Hurtado, 2009; Kramer, 2010; Hill et al., 2011; Sear & Coall, 2011; Meehan, Quinlan & Malcom, 2013). However, there is the issue that many adults appear to be primarily concerned with their own reproduction and that much of human behavior is clearly related to competitive breeding, including male–male competition, status striving, manipulation, and conflicts of interest even within families (Strassmann, 2011; Strassmann & Garrard, 2011). More caution is warranted in clearly determining the underlying benefits of actual individual behaviors. After all, are partible paternity practices best seen as cooperative breeding, mating competition, or both?

The greater female sexual autonomy and institutionalized extramarital relationships of partible paternity societies is predicted to generate reproductive conflict over the allocation of effort to parenting and mating. Men may respond to the opportunities for extramarital sex by increased mating effort, which trades off against their ability to deliver paternal care. In response to a dearth of parental effort on the part of mates, women may attempt to capitalize on these extramarital relationships by soliciting investment from multiple men named as fathers. Where relevant information is available, some amount of investment by secondary fathers (men named as co-fathers, but who are typically not the husband of the child’s mother) towards the mother and putative offspring has been noted for a number of partible paternity societies (e.g., Alès, 2002; Beckerman & Lizarralde, 2013; Beckerman & Valentine, 2002; Crocker, 2002; Hill & Hurtado, 1996; Kensinger, 2002). Among Barí horticulturalists of Colombia and Venezuela, unmarried women recruited greater numbers of secondary fathers for their children than married women (Beckerman & Lizarralde, 2013), suggesting a strategy aimed at maximizing male investment by women without a long-term mate. In the Ache hunter-gatherers of Paraguay, co-fathers were more likely to live together in the same band, as well as more likely to be related than men who were not co-fathers, suggesting that women chose co-fathers who were more likely and more able to invest in themselves and offspring (Ellsworth et al., 2014). In a milieu of unreliable paternal investment, provisioning and other forms of assistance by co-fathers could have important consequences for female reproductive success and child survival. Studies examining the effects of co-fathers among the Ache and Barí have shown that, where co-fathers invest in putative children and/or their mothers, this investment leads to higher rates of survival for children with multiple fathers (Hill & Hurtado, 1996; Beckerman et al., 1998; Beckerman & Valentine, 2002; Beckerman & Lizarralde, 2013; Ellsworth, 2014). That investment by secondary fathers drives this effect of increased survivorship of children with multiple fathers is supported by both Ache and Barí data where children with two fathers had the highest survival prospects, while children who did not have secondary fathers but whose siblings did, did not show increased odds of survival (Beckerman & Lizarralde, 2013; Hill & Hurtado, 1996).

Here we analyze co-father relationships among the Suruí of Brazil in order to test hypotheses on the reproductive strategies of men within the context of partible paternity (Walker, Flinn & Hill, 2010). Our focus on co-fathers is driven by a previous emphasis on mothers where partible paternity appears to make logical sense if women can garner investment from multiple mates, choosing co-fathers in ways that maximize the likelihood and amount of investment in themselves and their offspring (Ellsworth et al., 2014). With regard to men, the benefits are less obvious; why, for example, do men tolerate the risks of cuckoldry or provide investment in children who may not be their own? One hypothesized benefit to men of partible paternity is the establishment and strengthening of alliances or cooperative bonds between men who are co-fathers of the same children (male alliance hypothesis). This hypothesis predicts that co-fathers will have affiliative types of relationships such as being close relatives or friends. Another hypothesized benefit that men may derive from partible paternity is increased mating access to more females, and, by extension, greater chances of siring offspring with multiple females (mate competition hypothesis). In contrast to the male alliance hypothesis, this hypothesis does not predict cooperative or affiliative relations between men who share paternity or that they will be close kin. Tests of the aforementioned hypotheses permit insight into the nature of reproductive dynamics among the Suruí, as well as provide further evidence to bear on the question of cooperative breeding in traditional societies, because only the male alliance hypothesis is consistent with cooperative breeding, while the mate competition hypothesis is not.

Materials and Methods

Ethnographic background

The Suruí (endonym Paiter) are Tupi-Mondé speaking horticulturalists in the states of Rondônia and Mato Grosso, Brazil. The Suruí made first peaceful contact with outsiders in 1969. Today there are over 1,200 Suruí living in at least 12 villages with some that are far from one another, making intervillage visiting difficult, although some men do occasionally travel to distant villages. Suruí social structure has 4 exogamous patrilineal clans (Bontkes & Merrifield, 1985; Mindlin, 1991). Yvinec lived with the Suruí for 17 months in 2005–2007 and 2013 developing a genealogy and understanding of co-fathers and their social relationships. Fieldwork was conducted after informed consent and authorized by the Brazilian Minister of Science and Technology (MCT, portaria no 129 de 09/03/2005), the National Center for Research (CNPq, processo CMC 052/2004) and by the National Foundation of the Indian (FUNAI, no 25/CGEP/05, processo CMC 2905/04). The genealogy for the Suruí represents 75% of the total population in 2005 (Yvinec, 2011) and is available online at KinSources (http://kinsources.net). It contains 926 total individuals and 389 marriages that span approximately 7 generations. Suruí have a high level of polygyny with an average of 1.63 wives per married man. According to Yvinec’s (2011) latest count, the Kaban clan includes almost 50% of the whole population, the Ğamir about 30%, Ğamep 15%, while only a few members of the Makor clan are left.

With the ability to leverage at least some control over selecting mates, Suruí women likely had some latitude in the assignment of co-fathers for their offspring. The Suruí are tolerant of adulterous relationships only to a point, as husbands have been known to beat their wives if they hear about an affair. Suruí women are known to seek out attractive men as lovers, but they do not always choose fathers directly once the child is born as they have to deal with rumors and accusations from the fathers and others. When multiple fathers are from different clans, children are usually considered to belong to the clan of the primary father but debates about their clan membership often arise. The primary father, generally the man married to the mother, is assumed to be the genetic father in the genealogy.

There is no definitive statement of conception by the Suruí. The father is often said to “make most of the child,” the mother “only a little,” and some co-fathers are said to have made more than others. Fathers are said to transmit to their children through sexual conception some general skills associated with their clan such as being a good warrior or shaman. We know of 53 individuals with multiple fathers (only about 6% of the total population); of these, 47 individuals had 2 co-fathers and 6 had 3 co-fathers. In only 1 case was the identity of a co-father unknown, which yields 64 total co-father dyads.

Co-fatherhood among the Suruí refers to at least 3 different situations as observed by Yvinec. (1) A man has a wife who has an affair with another man who gets her pregnant; the husband keeps the wife and raises the child. (2) A man has a wife, but during early pregnancy she has an affair (late pregnancy sexual activity is prohibited in theory), and the husband keeps the wife and raises the child. (3) A man has a wife, but during pregnancy another man “takes” the wife and raises the child or the wife can be “given” by her first husband to the second. The Suruí mention that an elder brother or a father “lent” or “gave” a wife to a younger brother or son because the latter lacked a spouse. The identity of co-fathers and the attribution of primary versus secondary father may be well known to everybody, including the child, or can only be rumored and refused by the child. The co-fatherhood of an individual can be evoked in quite different ways, sometimes in a humorous way in his or her presence or in a pejorative way behind his or her back.

For most (40 of 64) co-father dyads there is no information on their social relationships because they died long ago or were little known to Yvinec. For 24 of the co-father dyads, it was straightforward to assess the qualitative nature of the relationship. Co-father relationships were organized into one of the following categories: (1) amicable (“got along,” such as men who are political allies, friends, or live together), (2) avoidant (e.g., some men moved villages because of a dispute), or (3) openly hostile (“did not get along,” such as one man who threatened to kill a co-father and another who requested a sorcery assault). The latter two categories are often directly related to jealousy over sexual relationships. In one notable dispute, a man was shot at by a distant cousin of another clan as a threat because of an adulterous affair (both were later named co-fathers); the threatened man was then given a wife by his father to put an end to the adultery and avoid more fighting.

Data analysis

To calculate relationships between co-fatherhood, genetic relatedness, and clan membership, three square similarity matrices were calculated for the 446 total men in the genealogy. Data on co-fathers includes all known co-father dyads in the Suruí population (n = 64). A co-fatherhood matrix codes all co-father pairs as 1 and all other pairs as 0. Clanship was coded in a similar fashion with 1 as pertaining to the same clan and 0 otherwise. A genetic relatedness matrix was calculated using Hagen’s Descent software (http://code.google.com/p/descent) which uses formulas from (Wiggans, Van Raden & Zuurbier, 1995). For our analyses, we used multiple regression on distance matrices (MRM, using the ecodist package in R; Goslee & Urban, 2007). For regression coefficients, MRM uses permutation tests of significance, and for the following analyses, we used 10,000 permutations per model. First, single predictor models were used to assess the relationships between all three matrices. Next, we regressed co-fatherhood on clanship and relatedness.

Results

Genetic relatedness of co-fathers

Average relatedness of the 64 Suruí co-father pairs is 0.129 (95% bootstrapped confidence interval 0.084–0.178), or around first cousin on average, and 61% are from the same clan. The average relatedness of all men alive recently is approximately a half-first cousin (0.057, 95% bootstrapped confidence interval 0.048–0.066). Therefore, average co-fathers are about twice as related as expected by chance. Figure 1A shows that co-fathers actually comprise slightly more unrelated (or low relatedness up to 0.01) dyads than expected by chance. Moreover, in the category of relatedness from 0.01 to 0.1, there are less co-fathers than expected by chance. In fact, the only category where co-fathers show higher relatedness than expected by chance is in the top category of 0.5 relatedness where 17% of all co-father dyads are father-son (n = 5) or full brothers (n = 6).

Figure 1 Frequency distribution of the relatedness between co-fathers for the Suruí (A) and Ache (B) with bootstrapped 95% confidence intervals as compared to random pairs of men.

Ache co-fathers are also about twice as related than expected by chance (r = 0.04 versus 0.02, Ellsworth et al., 2014, Fig. 1B). The Ache and Suruí genealogies are similar in size, quality, and depth. The primary difference is that the Suruí have a combination of more close kin marriages and higher polygyny which creates an intensive kinship network, while the Ache have few kin marriages and low polygyny which creates an extensive kinship network (Walker & Bailey, 2014; Bailey, Hill & Walker, 2014; Walker, 2014). In the Suruí, avunculate marriages between uncle and uterine niece are prescribed and cousin marriages are common; 20% of all Suruí marriages are between couples with at least first cousin relatedness (r ≥ 0.125), while this value is less than 1% for the Ache. Suruí have a high level of polygyny with an average of 1.63 wives per married man, whereas for the Ache it is 1.04 which creates many more paternal sibs in the Suruí. As illustrated in Fig. 1, the kin bias among co-fathers in the Ache emerges for kinship relationships with relatedness over 0.1, whereas for the Suruí the kin bias is only visible for closer kin with relatedness of 0.5 (i.e., brothers and father-son), perhaps because baseline genealogical relatedness is about 3 times higher in the Suruí.

Co-fatherhood, genetic relatedness, and clan membership

Results of the single predictor models showed that the relatedness matrix significantly predicted co-fatherhood (B = 0.012; p = 0.001). Clanship also predicted co-fatherhood (B = 0.009; p = 0.012). When co-fatherhood was regressed on both clanship and relatedness, the effect of relatedness remained significant (B = 0.011; p = 0.005), but the effect of clanship became borderline statistically significant (B = 0.006; p = 0.068), indicating co-father relatedness is not only a byproduct of co-fathers coming from the same clan.

Social relationships between co-fathers

Available information on co-father relationships, based on first-hand ethnographic observation, indicates that there are roughly two distinct categories of co-father relations: affiliative and avoidant/hostile (Table 1). In the affiliative category, co-father relationships are amicable, and occur among men of the same clan who are close kin such as brothers and father-son (n = 8 total with no exceptions). In the second category, relationships are of avoidance, competition, or open hostility, and are predominantly characteristic of co-fathers who are more distant kin (i.e., cousins including patrilateral, matrilateral, parallel, and cross, and uncle-nephew pairs, all brother’s son) and unrelated men of different clans (n = 13 total with 3 exceptions). Our sample of co-father dyads with known social relationships is small, but we have no reason to believe it is biased other than describing mostly recent social relationships.

Table 1 Social relationships between co-fathers of different relatedness categories.

Relatedness category	Affiliative	Avoidant or hostile	Unknown relationship	
Unrelated	1	3	28	
Cousins and Uncle-Nephew	2	10	6	
Brothers and father-son	8	0	6	

Discussion

The results of our analyses of Suruí co-father relationships show that shared paternity occurs between both close kin, as well as more distantly or unrelated men, and that the nature of the social relations between co-father dyads maps onto patterns of relatedness. Co-father relationships are amicable when they are between close kin but are more likely to be avoidant or openly hostile when they are between more distant kin or men of different clans. Results therefore imply plurality in men’s mating strategies, both cooperative and competitive, underlying partible paternity practices by Suruí men. Some Suruí men share parentage as a form of mate or wife giving, while others appear to poach on one another for access to more mates.

While we do not have a way to systematically estimate the base rate of different relationships among all men, we surmise that brothers and father-sons generally get along even if they are not co-fathers, especially when they live close to one another. The Suruí have an explicit ideology of solidarity between father and son and between brothers which likely suppresses the expression of jealousy when they are co-fathers. We also surmise that sexual jealously between cousins and unrelated co-fathers likely makes them hostile or to avoid each other. In general, cousin and uncle-nephew relationships are not known by the Suruí to be problematic relationships and are instead expected to show at least some solidarity, so it is notable that 10 of 12 were either hostile or avoidant when between co-fathers.

We suggest that in a society organized around patrilineal kinship and clan exogamy like the Suruí, fatherhood shared between men of different clans is likely to be associated with relatively more co-father hostility or avoidance, as it may lead to dispute over a child’s social identity (Peluso & Boster, 2002; Kensinger, 2002). This suggestion is supported by evidence from other patrilineal partible paternity societies wherein shared paternity among agnatic kin and fellow clansmen are tolerated while fatherhood shared between members of different clans are sources of conflict (e.g., Wanano Chernela, 2002, Curripaco Valentine, 2002). A number of cousins who are co-fathers in our Suruí sample are members of different clans, and it is therefore unsurprising that they fall into the general category of hostile or avoidant relations.

Examples of close kin, often brothers, sharing paternity appear widespread in other paternity partible societies, including the Curripaco (Valentine, 2002; Cormier, 2003; Erikson, 2002; Chernela, 2002; Alès, 2002). Formal friendship ties also exist between co-fathers in the Araweté (Viveiros de Castro, 1992), Canela (Crocker, 2002), and Arara (Walker, Flinn & Hill, 2010), appearing to support the male alliance hypothesis. In a previous study, our assumption was that most Ache co-fathers that were of first or second cousin relatedness or higher had amicable relationships (Ellsworth et al., 2014). However, the present results from the Suruí suggest that most cousins and even uncles and nephews have hostile or avoidant relationships. The Ache also appear to have two categories of co-father relations with some that tended not to like one another and were traditionally enemies at club fights. Some Ache men mentioned that they wanted to club other men who had sex with their wives and that some co-fathers were generally despised. Ache men with more primary fatherhood also have more secondary fatherhood. Whether or not this is because the man’s mate value causes opportunities for more fatherhood, a competition-based model best explains this result. That said, some co-father relations among the Ache were affiliative in nature as evidenced by higher levels relatedness and higher probability of co-residence, consistent with a male alliance hypothesis.

Partible paternity in the Ache and Suruí (and likely many other societies) offers good examples for why we should exercise caution in labeling humans as cooperative breeders based simply on certain cultural features. There is the nuance that most partible paternity behaviors from the women’s perspective may in fact be cooperative or communal breeding but would seem to be more variable from the perspective of men. As we have documented here, cooperative breeding would be applicable only to the affiliative co-fathers and potentially explainable by kin selection, while other instances of shared fatherhood may be best described as a form of male–male competition leading to hostile relationships between co-fathers. Men competing with one another for mates are clearly not sacrificing their own reproduction to invest in other men’s children, and may trade off some degree of cuckoldry risk for more investment in mating effort.

In conclusion, our study supports divergent strategies regarding the benefits of partible paternity. Our results cannot reject either the male alliance hypothesis or the mate competition hypothesis given that some co-father dyads are between closely related men with amicable relations while for others the relationship is hostile. These inherent complexities of partible paternity institutions add to our knowledge of the full range of cross-cultural variation in human mating and marriage tactics. They also show how the same cultural trait of partible paternity simultaneously includes aspects of both competitive and cooperative breeding.

This paper benefited from discussions with Kim Hill and Mark Flinn.

Additional Information and Declarations

Competing Interests

Author Contributions

Human Ethics

The authors declare there are no competing interests.

Robert S. Walker conceived and designed the experiments, performed the experiments, analyzed the data, wrote the paper, prepared figures and/or tables, reviewed drafts of the paper.

Cédric Yvinec performed the experiments, contributed reagents/materials/analysis tools, wrote the paper, reviewed drafts of the paper.

Ryan M. Ellsworth wrote the paper, reviewed drafts of the paper.

Drew H. Bailey analyzed the data, wrote the paper, reviewed drafts of the paper.

The following information was supplied relating to ethical approvals (i.e., approving body and any reference numbers):

Fieldwork among the Suruí was authorized in Brazil by the Minister of Science and Technology (MCT, portaria no 129 de 09/03/2005), the National Center for Research (CNPq, processo CMC 052/2004) and by the National Foundation of the Indian (FUNAI, autorisação no 25/CGEP/05, processo CMC 2905/04).

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
