# Peer review of "Co-father relationships among the Suruí (Paiter) of Brazil"

_PeerJ, doi:10.7717/peerj.899_

## Round 0.1 · original submission · Major Revisions

Two reviewers have provided thoughtful feedback on your manuscript. There appear to be some accuracy issues with the data reported, and the organization of the manuscript's content could use some rethinking as well. In addition to the reviewers' comments, you might want to consider these points as well:

(1) It is unclear from the text whether the ethnographic background (Lines 101-157) is drawn from the authors' own observations. Currently the section reads conclusory, and the location of discussion for ethical and/or regulatory approvals for the research is awkward. For readers unfamiliar with ethnographic methods, it would be helpful to explain and illustrate how you (Yvinec) reach the conclusions you recount there. For example, how were co-father relationships categorized (Lines 148-152)? Was it by one observer or multiple observers? Was this done qualitatively or perhaps quantitatively? Additional details would be helpful.

(2) Your thesis seems to be buried at Lines 82-83, no? However, the paragraph that follows seems under-developed (and causes the discussion particularly in Lines 252-260 to be a bit confusing and in Lines 262-268 to fall flat). If those hypotheses are the focus of your research and manuscript, they could be set out more clearly and captivatingly so that the rest of the manuscript flows more logically for the readers. In its current state, the text may confuse readers about whether those are your hypotheses here, common theories among scholars in the discipline, or both.

(3) In a few locations of the results (e.g., Lines 218-225) you make some guesses about what is going on. This probably is better suited for a discussion section, and the work might be more effective if you could provide some insight as to how your suspicions could be further investigated.

Additional details about your framing of the issues and a reorganization of the material could improve the basic reporting and facilitate the assessment of the validity of the results you found and conclusions you reach.

Reviewer 1 ·

Basic reporting

No comments.

Experimental design

Lines 137-38 say that there are 6 individuals with 3 co-fathers and 47 with 2 co-fathers. Assuming that all these co fathers are different me, that gives 47 + 18 = 65 co-father dyads. However, line 152 says there are 64 co-father dyads. Presumably there is a duplication somewhere, and it needs to be noted and explained.

Validity of the findings

The sample sizes for the (line 221) "roughly two separate categories of co-fathers" cross tabulated with amicable vs avoidant relations in Tab. 1 are quite small. A justification is needed for accepting the results.
More fine-grained attention to the ins and outs of the cooperative breeding issue would be helpful. This paper is about whether co-fathers count as cooperative breeders, and the conclusion, some do and some don't, is interesting and important. However, there are many ways in which humans may be cooperative breeders (e.g., aid from siblings,from previous children and from grandmothers) and it is not rights to jump from pointing out that (some) co-fathers are not cooperative breeders to the statement that humans are not cooperative breeders. The authors do not quite say this outright, but they come close (lines 262-266); a bit more caution might be appropriate.

Additional comments

Nice piece. Needs one more copy editing--check lines 89-90; 127-128, 190, 194-195.

·

Basic reporting

Please see general comments.

Experimental design

NA

Validity of the findings

Please see general comments.

Additional comments

This paper describes relationships among co-fathers in the Suruí of Brazil. It’s interesting to read about. I have several suggestions for the manuscript.

Most fundamentally, I am concerned with what kind of creature this manuscript is supposed to be. It’s very ambiguous.

The key statement in the abstract ‘We show that co-fathers roughly assort into two categories’ gives the impression that something more has been done than describe how close kin are more likely to get along with each other than distant kin. At least if something more than this has been shown, it is not at all clear to me what that might be.

More generally, there is no statement of what the paper is meant to be trying to achieve before the Results section arrives. It rather gives the impression that some basic descriptive results are being forced into a hypothesis-testing format. In fact, as far as I can tell, the first three paragraphs under the Results heading open with three descriptions of essentially the same result!

There are other things that bother me more generally about the way the parenting relationships in the Suruí (and indeed in societies in which partible paternity beliefs are present) are framed and discussed. I think that more thought is needed and more detail about what’s actually going on in this community. I think that the best way to improve the manuscript would be to make it more descriptive, ignoring the convention of methods, results, discussion, etc. This is a weak paper in that constrained format, but it could potentially be a strong one if it were more descriptive, ethnographic even.

Regarding the theoretical location of the subject, I think the understanding of what cooperative breeding is and where humans might fit into this is wrong. I recommend the below papers by Lukas and Clutton-Brock (2012) and Bogin et al (2014) that consider human reproduction in comparative context, and make what I think is a persuasive case for humans not being co-operative breeders in the strict sense. I think that if we’re discussing whether or not humans are co-operative breeders, Bogin et al’s breakdown in the early part of their paper of what Lukas and Clutton-Brock’s analysis means for primates (and thus humans) is really important, even if you don’t read the rest of either Bogin or Lukas in any detail. The word ‘co-operative’ here is perhaps something of a misnomer, something being lost in translation from this phrase being used first in animals and then in humans, but it’s a word we’re stuck with now.

Some important background context is lacking. It would be useful background to know what might be the costs and benefits of investing/not investing in offspring in the Suruí. Is there any strong evidence of costly investment by fathers or co-fathers or is it merely assumed? What form might that investment take?

I really want to know how accurate the information about co-father relationships might be, and how such confidence might be arrived at. Also, what determines whether a child is assigned to multiple fathers? Is there any chance that some children might attract multiple fathers because of the possibility of direct benefits to the co-father, such as greater ties to the child’s family? I think that this needs to be acknowledged as a possibility.

I know this stuff is difficult to measure, but it forms the backbone of the central premise upon which this kind of work pivots, that being a co-father is costly, either from a point of view of giving investment to a child, or because of cuckoldry. It shouldn’t be taken for granted, even if it seems obvious – in fact, especially if it seems obvious. To what extent is cuckoldry a useful concept here?

The term ‘secondary fathers’ needs to be defined.

The degree of polygyny would ideally be mentioned earlier, and make reference to the characteristics of unmarried men (e.g. age, productivity, etc.)

Concerning the phrasing in the paragraph beginning on p.134, is the man ‘having’, ‘keeping’, ‘giving’ a wife a pattern of description that is emic or etic in origin? I imagine it’s quite a grounded description, but if so it would benefit from being clearer.

Is comparing the relatedness of men in co-father relationships to ‘random’ men justified, as opposed to comparing them to men who are more likely to associate with one another anyway?

Hope that this input is useful.


Ian Rickard
Dept. Anthropology, Durham University




Refs:

Bogin B, Bragg J, Kuzawa C (2014) Humans are not cooperative breeders but practice biocultural reproduction. Ann Hum Biol 41:368–380.

Lukas D, Clutton-Brock TCH (2012) Cooperative breeding and monogamy in mammalian societies. Proceedings of the Royal Society B: Biological Sciences 279:2151–2156.

---

## Round 0.2 · accepted · Accept

Thank you for addressing the reviewers' concerns with the original manuscript. There is one minor correction for basic reporting that should be made (replace "she" with "women" in the sentence on lines 98-101).

Reviewer 1 ·

Basic reporting

This version of the manuscript addresses the problems the reviewers found in the first draft, and in my opinion is now ready to be published. My only complaint is the sentence (lines 98-101): “Our focus on co-fathers is driven by a previous emphasis on mothers where partible paternity appears to make logical sense if she can garner investment from multiple mates, choosing co-fathers in ways that maximize the likelihood and amount of investment in themselves and their offspring (Ellsworth et al., 2014).” It should read: “Our focus on co-fathers is driven by a previous emphasis on mothers where partible paternity appears to make logical sense if women can garner investment from multiple mates, choosing co-fathers in ways that maximize the likelihood and amount of investment in themselves and their offspring (Ellsworth et al., 2014).”

Experimental design

No problems.

Validity of the findings

Interesting and important.